# In-depth Analysis of the Lid Subunits Assembly Mechanism in Mammals

**DOI:** 10.3390/biom9060213

**Published:** 2019-05-31

**Authors:** Minghui Bai, Xian Zhao, Kazutaka Sahara, Yuki Ohte, Yuko Hirano, Takeumi Kaneko, Hideki Yashiroda, Shigeo Murata

**Affiliations:** Laboratory of Protein Metabolism, Graduate School of Pharmaceutical Sciences, The University of Tokyo, 7-3-1 Hongo, Bunkyo-ku, Tokyo 113-0033, Japan; hakumeikei@gmail.com (M.B.); zhao-xian762@g.ecc.u-tokyo.ac.jp (X.Z.); kztkshr@gmail.com (K.S.); yuki.ohte@gmail.com (Y.O.); yuuko92130@yahoo.co.jp (Y.H.); takeumi@gmail.com (T.K.); yashiroda@mol.f.u-tokyo.ac.jp (H.Y.)

**Keywords:** 26S proteasome, 19S regulatory particle, lid subcomplex, Rpn proteins, assembly

## Abstract

The 26S proteasome is a key player in the degradation of ubiquitinated proteins, comprising a 20S core particle (CP) and a 19S regulatory particle (RP). The RP is further divided into base and lid subcomplexes, which are assembled independently from each other. We have previously demonstrated the assembly pathway of the CP and the base by observing assembly intermediates resulting from knockdowns of each proteasome subunit and the assembly chaperones. In this study, we examine the assembly pathway of the mammalian lid, which remains to be elucidated. We show that the lid assembly pathway is conserved between humans and yeast. The final step is the incorporation of Rpn12 into the assembly intermediate consisting of two modular complexes, Rpn3-7-15 and Rpn5-6-8-9-11, in both humans and yeast. Furthermore, we dissect the assembly pathways of the two modular complexes by the knockdown of each lid subunit.

## 1. Introduction

Protein degradation exerted by the ubiquitin–proteasome system starts with the conjugation of ubiquitin chains to target proteins. Polyubiquitinated proteins are then recognized and captured by the 26S proteasome and digested to short peptide fragments [1]. The ubiquitin–proteasome system is required for various cellular processes such as DNA damage repair, cell cycle progression, signal transduction, and immune response [2].

The 26S proteasome is made up of the 20S core particle (CP) and the 19S regulatory particle (RP) [3]. As a prerequisite for protein degradation by the CP, substrate proteins need to be recruited, deubiquitinated, and unfolded by the RP [4]. The RP can be further divided into a “base” and a “lid” subcomplex. The base subcomplex is composed of six ATPase subunits (Rpt1–Rpt6) and three non-ATPase subunits (Rpn1, Rpn2, and Rpn13), whereas the lid subcomplex is composed of nine non-ATPase subunits (Rpn3, Rpn5–Rpn9, Rpn11, Rpn12, and Rpn15/Sem1), and Rpn10 appears to mainly bind the lid subunits [5,6,7,8].

The assembly of the 26S proteasome is highly complex because of the large number of subunits that require precise association to form an active complex. Previous studies have demonstrated assembly pathways of the CP and the base subcomplex. CP biogenesis, consisting of seven different α-type subunits (α1–α7) and seven different β-type subunits (β1–β7), requires the assistance of at least five proteasome-specific chaperones called PAC1–4/Pba1–4 and POMP/Ump1 in mammals/yeast [9,10,11,12,13,14,15,16]. The assembly of the base subcomplex involves another set of chaperones, including p28/Nas6, p27/Nas2, S5b/Hsm5, and PAAF1/Rpn14 [17,18]. In both the CP and the base, the order of assembly has been clarified in detail using the siRNA-mediated knockdown of each subunit [17,19].

Recent biochemical studies showed that the lid and the base form independently [17,20]. Studies using yeast mutants and the mass spectrometry analysis of native complexes suggested that the lid consists of two modules: a complex comprised of Rpn5, Rpn6, Rpn8, Rpn9, and Rpn11 and a complex comprised of Rpn3, Rpn7, and Rpn15/Sem1 (also called DSS1 in mammals; hereafter called Rpn15) [21,22,23]. These two modules seem to be connected via Rpn6 [22]. Rpn12 incorporation depends on the pre-assembly of all the other lid subunits and occurs as the last step of the lid assembly, which drives lid–base joining [24].

In addition, the molecular architecture of the 26S proteasome was resolved in detail using cryoelectron microscopy, which revealed the subunit topology of the lid [25,26,27]. The PCI domain-containing subunits Rpn3, Rpn7, Rpn6, Rpn5, and Rpn9 form the fingers of the hand-shaped lid structure. The PCI domain is thought to mediate protein–protein interactions. Rpn8 connects Rpn3 and Rpn9, and Rpn11 lies in the palm of the hand. This structure is consistent and would explain the results regarding lid assembly which have been reported so far. However, the questions of how each subunit assembles into the modules and how the modules assemble into the lid remain to be answered.

In this study, we investigated the biogenesis of the mammalian lid subcomplex using a combination of RNA interference and mass spectrometry. siRNA knockdown of each lid subunit caused a characteristic change in distributions of the other lid subunits separated by glycerol gradient centrifugation. We purified complexes from the separated fractions as lid intermediates by immunoprecipitation. The components of the intermediates were determined by mass spectrometry. These analyses demonstrated the assembly pathway of the two modules of the lid subcomplex, as well as the assembly pathway of the lid in detail.

## 2. Materials and Methods

### 2.1. Cell Culture and DNA Constructs

HEK293T cells were cultured at 37°C with 5% CO_2_ in Dulbecco’s modified Eagle’s medium (Nacalai Tesque, Kyoto, Japan) containing 10% fetal bovine serum (Invitrogen, Waltham, Massachusetts, USA) and 1% penicillin–streptomycin (Nacalai Tesque). The cDNAs encoding Rpn5, Rpn7, and Rpn9 were subcloned into the pIRESpuro3-Flag vector, and the cDNA encoding Rpn15 was subcloned into the pIRESpuro3-GFP vector. Transfection of these constructs into HEK293T cells was achieved by FuGENE 6 (Roche Applied Science, Penzberg, Germany), and the cells were selected with 4 µg/mL puromycin (InvivoGen, San Diego, California, USA).

### 2.2. RNA Interference

The siRNAs targeting human lid subunits (Table 1) were transfected into HEK293T cells using Lipofectamine RNAiMAX (Invitrogen) at a final concentration of 50 nM. For each sample, 9 × 10^5^ cells were plated in a 10-cm dish 6 h before transfection, and the cells were harvested 48 h after transfection.

### 2.3. Protein Extraction, Immunological Analysis, and Antibodies

Cells were lysed in lysis buffer [20 mM Tris-HCl (pH 7.5), 0.2% (*w/w*) NP-40, 1 mM dithiothreitol, 2 mM ATP, and 5 mM MgCl_2_] and incubated at 4°C for 10 min. The lysates were retrieved by centrifugation at 20,000 *g* for 10 min at 4°C and separated by 4%–24% glycerol gradient centrifugation as described previously [19,28]. Antibodies against human lid subunits, GFP, and Flag were used [17].

### 2.4. Immunoprecipitation and Mass Spectrometry

The accumulated intermediates were immunoprecipitated with M2 agarose (Sigma, St. Louis, Missouri, USA) as described previously [17]. The precipitated complexes were analyzed by LC-MALDI followed by tandem mass spectrometry using a TOF/TOF5800 analyzer (Applied Biosystems, Waltham, Massachusetts, USA).

### 2.5. In Vitro Transcription and Translation

In vitro labeling was performed using the TNT T7 Quick for PCR DNA system (Promega, Madison, Wisconsin, USA) with the EXPRESS Protein Labeling Mix [^35^S] (PerkinElmer, Waltham, Massachusetts, USA), according to the manufacturer’s protocol. The labeled samples were incubated with M2 agarose in a buffer containing 25 mM Tris-HCl (pH 7.5), 150 mM NaCl, and 0.2% NP-40. The washed beads were boiled in the SDS-sample buffer and separated by SDS-PAGE, followed by autoradiography visualizing.

## 3. Results

### 3.1. Interdependence Between the Lid Subunits for Protein Expression

To investigate the assembly pathway of the human lid subcomplex, we performed the knockdown of each lid subunit using siRNAs targeting Rpn3, Rpn5–Rpn9, Rpn11, Rpn12, and Rpn15 in HEK293T cells. The cells were harvested 48 h after siRNA transfection, when they were still viable. Further extension of any knockdown caused cell death, indicating that all the lid subunits are essential for viability in mammalian cells (data not shown). We examined the cell extracts by immunoblot analysis for each lid subunit, a base subunit Rpt6, and a CP subunit α6 (Figure 1A). Decreases in the targeted subunits were demonstrated, except for Rpn15, against which we failed to obtain an antibody (Figure 1A). The peptidase activity of the 26S proteasome in each knockdown showed a 30–50% reduction compared with control cells, which also indicates successful knockdown of the lid subunits (Figure 1B).

The protein levels of Rpn3 and Rpn5–Rpn8 were decreased only by their corresponding siRNAs, and the protein levels of Rpn9, Rpn11, and Rpn12 were also decreased by siRNAs targeting other lid subunits. Rpn5-knockdown caused a reduction in Rpn9 and Rpn11 as well as Rpn5. Rpn11 was also decreased by Rpn6-knockdown and Rpn8-knockdown (Figure 1A). Because it has been suggested that these five subunits, i.e., Rpn5, Rpn6, Rpn8, Rpn9, and Rpn11, form a modular structure before the lid assembly, these results indicate interdependence for protein stability within this module. A severe reduction of Rpn12 was observed in Rpn3-knockdown, and a moderate reduction of Rpn12 was observed in Rpn7- and Rpn15-knockdown. Because Rpn3, Rpn7, and Rpn15 are also known to form a modular structure before the lid assembly, these results suggest that the protein expression of Rpn12 is dependent on the presence of the Rpn3-7-15 module. No decrease of the other subunits was observed in Rpn11- and Rpn12-knockdown, suggesting that these two subunits are not responsible for the stability or expression of the other subunits (Figure 1A).

### 3.2. Mammalian Cells Express a Lid-like Complex Without Rpn12 (LP2)

We previously observed that mammalian cells contained an appreciable amount of complex containing lid subunits [17]. Lysates of HEK293T cells were separated by 4%–24% glycerol gradient centrifugation, followed by immunoblotting with antibodies for proteasome subunits (Figure 2A). Fraction 32 and Fraction 22 correspond to the peak location of the 26S proteasome (2.5 MDa) and the free CP (720 kDa), respectively, as indicated by the immunoblot analysis (Figure 2A). The α-ring (280 kDa), composed of the seven α-type subunits and PAC1–PAC2 and PAC3–PAC4 chaperone heterodimers, was also found around Fraction 12. In addition, all the lid subunits except Rpn12 had cosedimented in Fraction 12–14, suggesting that a subassembly of the lid exists in mammalian cells without any intervention (Figure 2A). Indeed, when we analyzed this complex by mass spectrometry utilizing HEK293T cells that expressed Rpn7-Flag, it turned out that this complex included all the lid subunits except Rpn12; however, we were not able to detect peptides derived from Rpn15, presumably because of the properties of the peptides (Figure 2B; see also Figure 5).

The same complex was reported as LP2 (lid particle 2) in yeast *rpn12* mutants [24,29]. This complex was capable of completing lid assembly once Rpn12 had been added, and for subsequent assembly of the 26S proteasome, showing that Rpn12 incorporation is the last step in the lid assembly of yeast [24]. Consistent with the observation in yeast, Rpn12-knockdown in HEK293T cells caused accumulation of the LP2 complex, which was further confirmed by mass spectrometry analysis of the complex (Figure 2B). At the same time, these cells accumulated aberrant complexes containing the base subunit Rpt6 in Fraction 8–10, Fraction 16–18, and Fraction 24–26, presumably corresponding to the Rpt3–Rpt6 module, the base subcomplex, and dimers of the base, respectively [17]. These results suggest that Rpn12 incorporation is the last step of lid assembly, after which the completed lid joins the base in mammalian cells.

### 3.3. Rpn6 is Required for Interaction Between Rpn3-7-15 and Rpn5-8-9-11 and for Rpn11 Stability

In the *rpn6* yeast mutant, a complex comprised of Rpn5, Rpn8, Rpn9, and Rpn11 did accumulate, suggesting that this complex is connected with other subunits via Rpn6 [22]. When Rpn6 was knocked down in HEK293T cells, two intermediates appeared in Fraction 6–10 (Figure 3). One corresponded to a complex of Rpn3 and Rpn7 and the other to a complex of Rpn5, Rpn8, Rpn9, and Rpn11. This result indicates that the two complexes, Rpn3-7-15 and Rpn5-8-9-11, can be formed independently and are connected to each other through Rpn6, similar to the situation shown in yeast [22]. Interestingly, Rpn6-knockdown caused a reduction in Rpn11 (Figure 1A). Consistent with this, the relative abundance of Rpn11 seems to be lower than that of the LP2 complex observed in wild-type cells and Rpn12-knockdown cells, suggesting that the association of Rpn6 to the Rpn5-8-9-11 complex increases the stability of Rpn11 (Figure 3).

### 3.4. Loss of Rpn11 does not Affect the Assembly of Other Lid Subunits but Affects Lid–base Joining

Of the lid subunits, Rpn11 is the only subunit that has well-known catalytic activity, and its deubiquitinating activity is essential for the degradation of ubiquitinated proteins by the proteasome. To examine the role of Rpn11 in the lid assembly, knockdown of Rpn11 was performed. Unexpectedly, even without Rpn11, other lid subunits except Rpn12 were able to assemble into a complex (Figure 4, Fraction 10–14). However, the lid–base joining was severely impaired as indicated by the accumulation of the base subcomplex in Fraction 16–18 and Fraction 20–24 (Figure 4; blot for Rpt6). This finding is consistent with the notion that Rpn12 is a checkpoint that monitors the integrity of the LP2 complex and suggests that the association of Rpn11 is essential for lid–base joining, presumably by supporting proper conformation of the LP2 complex.

### 3.5. Interdependent Assembly of the Rpn5-8-9 Complex

Rpn11-knockdown showed that Rpn11 is not an essential structural component in the formation of the Rpn5-8-9-11 complex. Therefore, we examined how the Rpn5-8-9 complex is assembled (Figure 5).

In Rpn5-, Rpn8-, or Rpn9-knockdown, the assembly of the Rpn3-7-15 module was not affected, and a small amount of Rpn6 was associated with the module, in favor of the Rpn3-7-15 module being assembled independently of the Rpn5-8-9-11 complex (Figure 5A–C). In Rpn5-knockdown, Rpn9 did not associate with Rpn8, as indicated by immunoprecipitation by Rpn9-Flag from the accumulated intermediates in Fraction 4–10 and mass spectrometry analysis of the resultant precipitates (Figure 5A). Through similar experiments, we found that Rpn5 did not associate with Rpn9 without Rpn8 (Figure 5B) and that Rpn5 did not associate with Rpn8 without Rpn9 (Figure 5C). These results indicate that Rpn5, Rpn8, and Rpn9 form a complex only when all the three subunits are present. This Rpn5-8-9 complex might serve as a core complex for the addition of the essential deubiquitylase Rpn11, because the loss of Rpn5 and Rpn8 significantly reduced the protein level of Rpn11 (Figure 1A).

### 3.6. Rpn7–Rpn6 Interaction Connects the Rpn3-7-15 Module with the Rpn5-6-8-9-11 Module

The Rpn3-7-15 module associates with the Rpn5-8-9-11 complex via Rpn6 (Figure 3). We next investigated how the Rpn3-7-15 module is assembled and how it binds to Rpn6.

When Rpn3 was knocked down, Rpn7-Flag in the intermediates in Fraction 10–14 coprecipitated with the Rpn5-6-8-9-11 module (Figure 6A). In contrast, without Rpn7, Rpn3-Flag in the intermediate fraction did not associate with the Rpn5-6-8-9-11 module, which was readily formed in the absence of Rpn7 (Figure 6B). When immunoprecipitated with Rpn5-Flag in Rpn7-knockdown cells, a small amount of Rpn3 was detected, suggesting that Rpn3 could directly associate with the Rpn5-6-8-9-11 module although Rpn7 is required for efficient incorporation of Rpn3 to the lid. Rpn15-knockdown exhibited essentially the same phenotype as Rpn3-knockdown; a complex was accumulated that consisted of Rpn7 and the Rpn5-6-8-9-11 module and contained only a small amount of Rpn3 (Figure 6C). Although Rpn5 was readily detected by mass spectrometry analysis of the complex in Rpn15-knockdown cells, it was not detected by immunoblot analysis owing to a potential protein modification (Figure 6C). These results suggest that the Rpn3-7-15 module interacts with Rpn6 via Rpn7, which connects the Rpn3-7-15 module with the Rpn5-6-8-9-11 module as a result. These data also suggest that Rpn15 is required for Rpn3 association with Rpn7.

### 3.7. Rpn15 Mediates the Association of Rpn3 with Rpn7

As we were unable to identify endogenous Rpn15 either by mass spectrometry or by immunoblot analysis, we generated HEK293T cells stably expressing Rpn15-GFP to investigate the role of Rpn15 in the assembly of the lid subcomplex.

Knockdowns of Rpn7, Rpn3, and Rpn6 were performed in the Rpn15-GFP-expressing cells, and the cell lysates were fractionated by glycerol gradient centrifugation. Immunoblot analysis of each fraction essentially showed the same results as knockdown in normal HEK293T cells (data not shown; refer to Figure 2A, Figure 3 and Figure 6A,B). The normal LP2 complex of control cells (corresponding to Fraction 12–16 of Figure 2A), accumulated lid intermediates of knockdown cells (corresponding to Fraction 6–14 of Figure 3 and Figure 6A,B; referred to as “Lid” in Figure 7), and 26S proteasomes (corresponding to Fraction 32 of Figure 2A, Figure 3, and Figure 6A,B; referred to as “26S” in Figure 7) of knockdown or control cells were immunoprecipitated with anti-GFP antibody.

Subunits comprising the LP2 complex were coprecipitated from control lid fractions as well as 26S fractions with Rpn15-GFP, indicating that Rpn15 is a component of the LP2 complex (Figure 7, lanes 9, 10, 12, 14, and 16). When Rpn6 was knocked down, Rpn15 was coprecipitated with Rpn3 and Rpn7 but not with the Rpn5-6-8-9-11 module, confirming that Rpn15 is indeed a component of the Rpn3-7-15 module. However, Rpn15 associated only with Rpn3 in the absence of Rpn7 (Figure 7, lane 11) and did not associate with any other subunits in the absence of Rpn3 (Figure 7, lane 13). These results suggest that (i) the assembly of the Rnp3-7-15 module starts with Rpn3 and Rpn15 forming a complex and (ii) Rpn7 subsequent recruitment is dependent on Rpn15.

### 3.8. Rpn15 Directly Binds to Rpn3 and Promotes Rpn3–Rpn7 Association in Vitro

To further clarify how the Rpn3-7-15 module is assembled, Flag-Rpn3, Rpn7, and GFP-Rpn15 were cotranscribed/cotranslated in vitro in various combinations (Figure 8). Flag-Rpn3 pulled down GFP-Rpn15 in the absence of Rpn7 and did not pull down Rpn7 without Rpn15. This result indicates that Rpn3 directly interacts with Rpn15 and that Rpn3 and Rpn7 either do not interact or exhibit very weak interaction with each other (Figure 8A). However, Flag-Rpn3 pulled down Rpn7 in the presence of GFP-Rpn15, indicating that Rpn15 is required for the association between Rpn3 and Rpn7.

Similar results were observed by mixing Rpn3, Rpn7, and Flag-Rpn15 (Figure 8B). Flag-Rpn15 directly interacted with Rpn3 and pulled down Rpn7 only when Rpn3 was present. The binding of Rpn15 to Rpn3 might change the Rpn3 and/or Rpn15 conformation so that Rpn3 and/or Rpn15 can interact with Rpn7.

## 4. Discussion

Previous studies from yeast *Saccharomyces cerevisiae* have demonstrated that the assembly of the proteasome lid begins with the formation of two modular complexes: Rpn3-7-15 and Rpn5-6-8-9-11 [24,29,30]. These two modules are assembled, followed by the incorporation of Rpn12 that completes the lid assembly and drives lid–base joining. 

In this study, we investigated the mammalian lid assembly pathway by observing complexes resulted from siRNA knockdown of each lid subunit and how the two modular complexes are formed. Although we cannot exclude the possibility that the intermediates we purified by immunoprecipitation are non-physiological intermediates such as those with abnormal subunit stoichiometries and dead-end products, we can describe the mammalian assembly pathway of the lid that conforms to the yeast model (Figure 9). Formation of the Rpn3-7-15 module starts with the association of Rpn15 and Rpn3, which enables Rpn7 incorporation. The Rpn5-6-8-9-11 module can be further divided into a Rpn5-8-9 complex, Rpn6, and Rpn11 on the basis of interdependence between the subunits. Whereas the *rpn11-1* mutation inhibits formation of the Rpn5-6-8-9-11 module and leads to accumulation of the Rpn3-7-15 module in *S. cerevisiae* [24], knockdown of Rpn11 in human cells indicates that mammalian Rpn11 is not required for the stability of the other lid subunits and can be incorporated even at the last step of LP2 formation. The minimal core of this module seems to be Rpn5-8-9, which can accommodate Rpn11. The association of Rpn6 to the core increases the capacity to accommodate Rpn11. The Rpn3-7-15 module and the Rpn5-6-8-9-11 module are connected between Rpn6 and Rpn7, forming the LP2 complex. Finally, Rpn12 is incorporated to complete lid assembly. 

In our experiments, Rpn15 directly bound only to Rpn3, as reported previously [31]. Rpn15 did not bind to Rpn7 when Rpn3 was absent, and Rpn7 and Rpn3 did not interact with each other without Rpn15. Thus, we speculate that Rpn15 might interact first with Rpn3, which leads to potential conformational change in Rpn15 and/or Rpn3 to assist the incorporation of Rpn7 to the intermediate of Rpn3-7-15. On the other hand, it is reported that Sem1, yeast ortholog of Rpn15, can bind to both Rpn3 and Rpn7 independently via its N- and C-termini, and tethers together Rpn3 and Rpn7 [32]. Furthermore, Sem1 and Rpn15 are positioned between Rpn7 and Rpn3 in the cryoelectron microscopy structures of the 26S proteasome [27,33]. Considering these preceding studies, it seems reasonable to suppose that Rpn15 also interacts with Rpn7 after binding to Rpn3, although our results do not necessarily require the direct interaction of Rpn15 to Rpn7. 

As for another interesting issue regarding Rpn15, less accumulation of the base subunit Rpt6 was observed in Rpn15-knockdown cells than other lid subunits (Figure 6C). Despite the abnormality in the formation of the lid, accumulation of the base was also affected by the knockdown of Rpn15, which needs to be further clarified not only for the assembly mechanism but also at the transcriptional level.

Recent research shows that overexpression of the Rpn6 subunit in *Caenorhabditis elegans* and of Rpn11 in the fruit fly prolongs their lifespan [34]. In addition, the FOXO4-mediated upregulation of Rpn6 is a prerequisite for maintaining pluripotency in embryonic stem cells [35]. The overexpression of Rpn11 or Rpn7 is involved in DNA damage response [36,37]. However, the mechanisms are still unidentified, and elucidating the assembly pathway of the lid subcomplex under these circumstances with overexpressed lid subunits might help in understanding the underlying mechanisms. 

Whether specific chaperones are involved in the mammalian lid assembly remains uncertain. We detected some proteins along with proteasome subunits in our mass spectrometry analysis (data not shown). We have not yet been able to discriminate whether these proteins are merely contaminated sediments or have potential biological importance. Although the lid can be reconstituted in *Escherichia coli* only with the expression of the yeast lid subunits [26,32], it may be possible to identify lid assembly chaperones among these proteins in future. Finally, the lid and the base, which are both assembled independently, are associated with each other to form the complete 19S RP. Further studies are needed to clarify the mechanism and the timing of the lid–base association.

## 5. Conclusions

This study describes the assembly pathway of the mammalian lid subcomplex of the RP in the 26S proteasome. The lid formation consists of the assemblies of two modular complexes: Rpn3-7-15 and Rpn5-6-8-9-11. The binding of Rpn15 to Rpn3 enables Rpn3 to interact with Rpn7. In Rpn5-6-8-9-11 formation, Rpn5-8-9 serves as a platform for the association of Rpn6 and Rpn11. Although Rpn11 binds to Rpn5-8-9 without Rpn6, Rpn6 has an effect on Rpn11 stability. The interaction between Rpn6 and Rpn7 puts Rpn3-7-15 and Rpn5-6-8-9-11 together. Rpn12 is the last subunit to be incorporated into the lid intermediate comprising all the other lid subunits, known as LP2. 

## Figures and Tables

**Figure 1 biomolecules-09-00213-f001:**
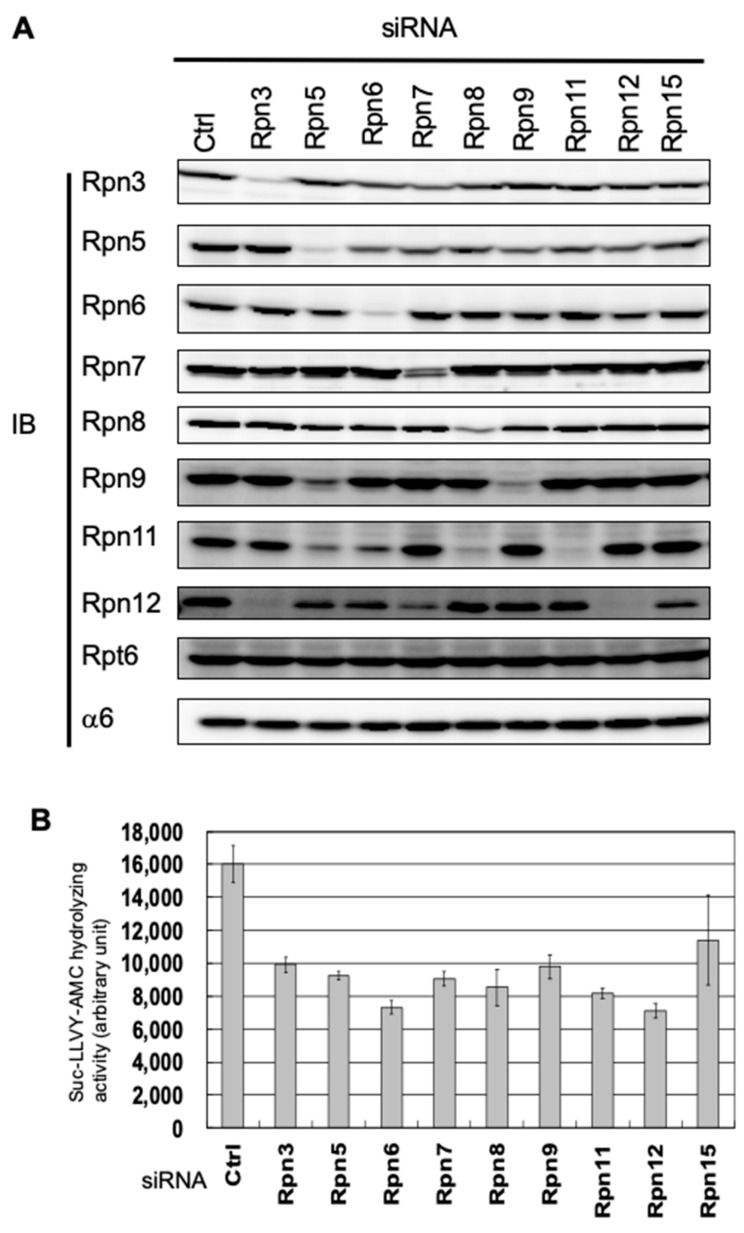
Interdependence between the lid subunits for protein expression. (**A**) siRNAs targeting Rpn3, Rpn5–Rpn9, Rpn11, Rpn12, and Rpn15 were transfected into HEK293T cells. The cell extracts (20 μg) were then separated by native PAGE and were detected by immunoblot (IB) using antibodies against Rpn3, Rpn5–Rpn9, Rpn11, Rpn12, Rpt6, and α6. (**B**) The Suc-LLVY-AMC hydrolyzing activity of the knockdown of Rpn3, Rpn5–Rpn9, Rpn11, Rpn12, and Rpn15 was measured without added SDS (mean ± SD, *n* = 3).

**Figure 2 biomolecules-09-00213-f002:**
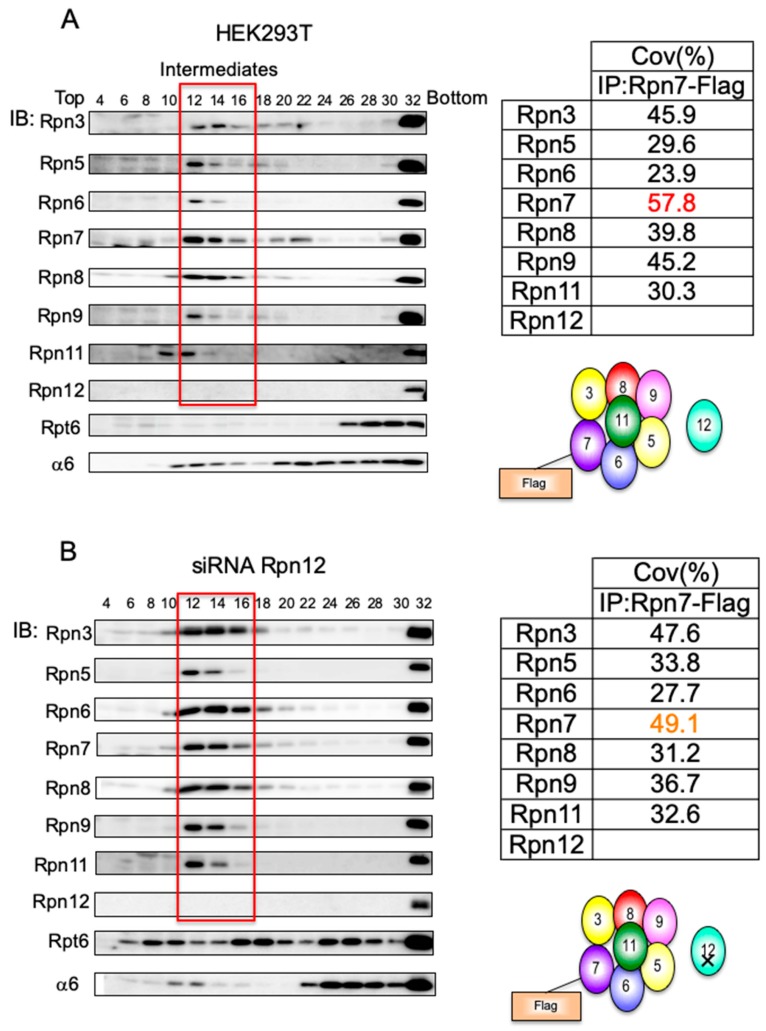
Rpn12 is incorporated at the final step of lid formation. (**A**) HEK293T cells stably expressing Rpn7-Flag were lysed and subjected to 4%–24% glycerol gradient centrifugation. The fractions were analyzed by immunoblotting with anti-Rpn3, Rpn5–Rpn9, Rpn11, Rpn12, Rpt6, and α6 antibodies. Fractions corresponding to the accumulated complexes (boxed in red) were immunoprecipitated (IP) with the anti-Flag antibody. The precipitated complexes were analyzed by LC-MALDI followed by tandem mass spectrometry. The table shows the sequence coverage of the identified subunits. (**B**) siRNA targeting Rpn12 was transfected into HEK293T cells for 48 h and then analyzed as described in Figure 2A.

**Figure 3 biomolecules-09-00213-f003:**
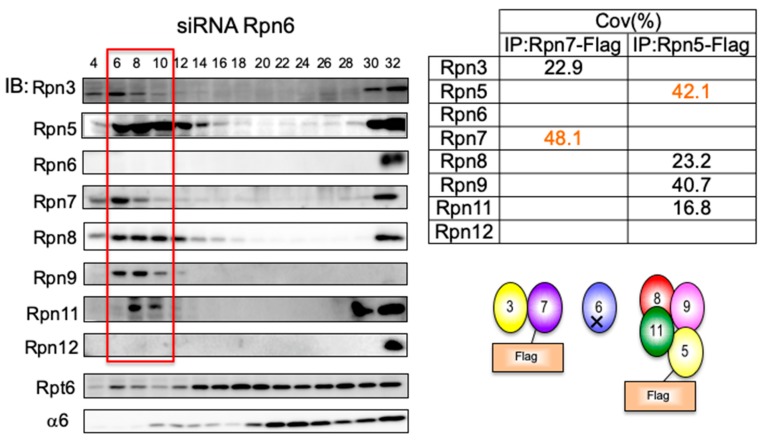
Rpn6 is required for the interaction between Rpn3-7-15 and Rpn5-8-9-11, and for Rpn11 stability. HEK293T cells stably expressing Rpn5-Flag and Rpn7-Flag treated with siRNA targeting Rpn6 for 48 h were analyzed in the same way as described in Figure 2. The table shows the sequence coverage of the identified subunits.

**Figure 4 biomolecules-09-00213-f004:**
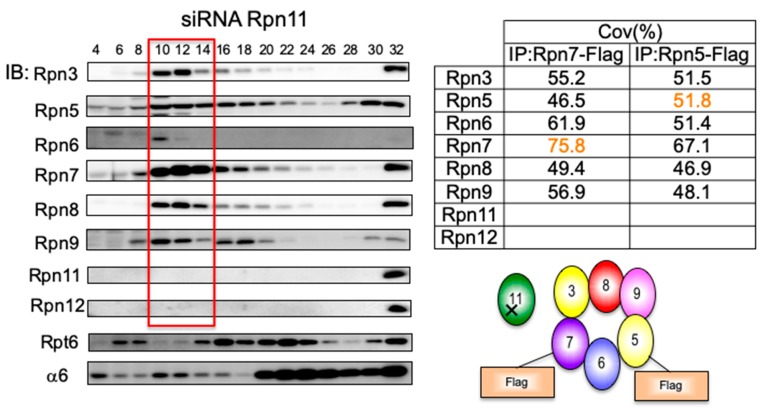
Loss of Rpn11 does not affect the assembly of other lid subunits but affects lid–base joining. HEK293T cells stably expressing Rpn5-Flag and Rpn7-Flag treated with siRNA targeting Rpn11 were analyzed for 48 h in the same way as described in Figure 2. The table shows the sequence coverage of the identified subunits.

**Figure 5 biomolecules-09-00213-f005:**
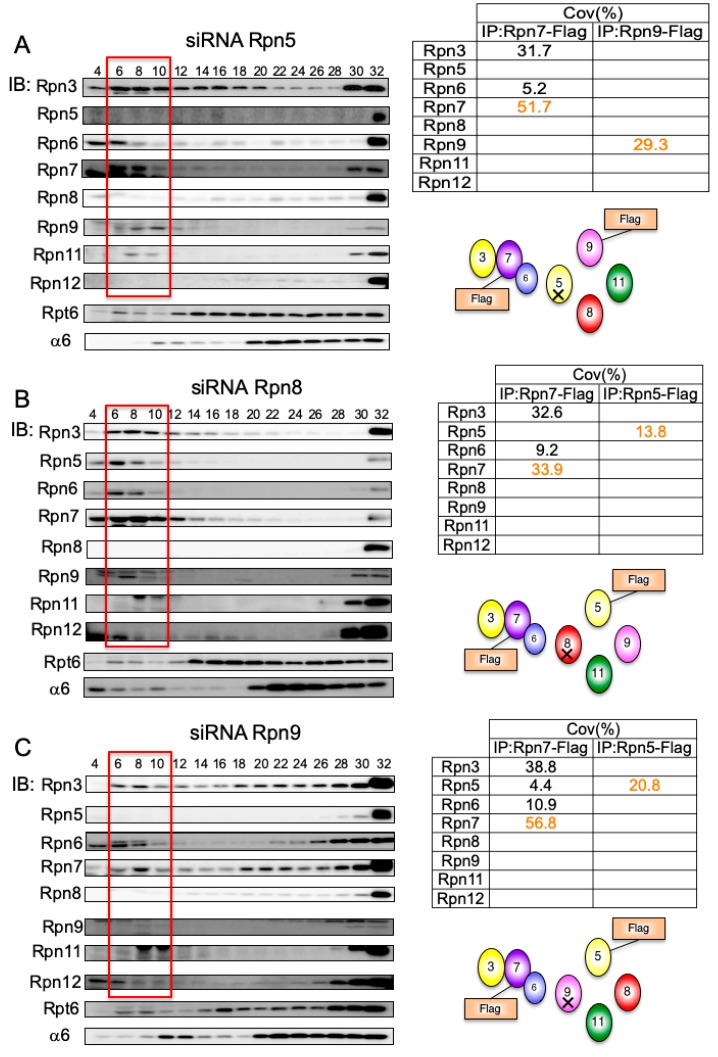
Interdependent assembly of the Rpn5-8-9 complex. HEK293T cells stably expressing Rpn7-Flag and Rpn9-Flag treated with siRNA targeting Rpn5 (**A**), cells stably expressing Rpn5-Flag and Rpn7-Flag treated with siRNA targeting Rpn8 (**B**), and cells stably expressing Rpn5-Flag and Rpn7-Flag treated with siRNA targeting Rpn9 (**C**) were analyzed in the same way as described in Figure 2. The tables show the sequence coverage of the identified subunits.

**Figure 6 biomolecules-09-00213-f006:**
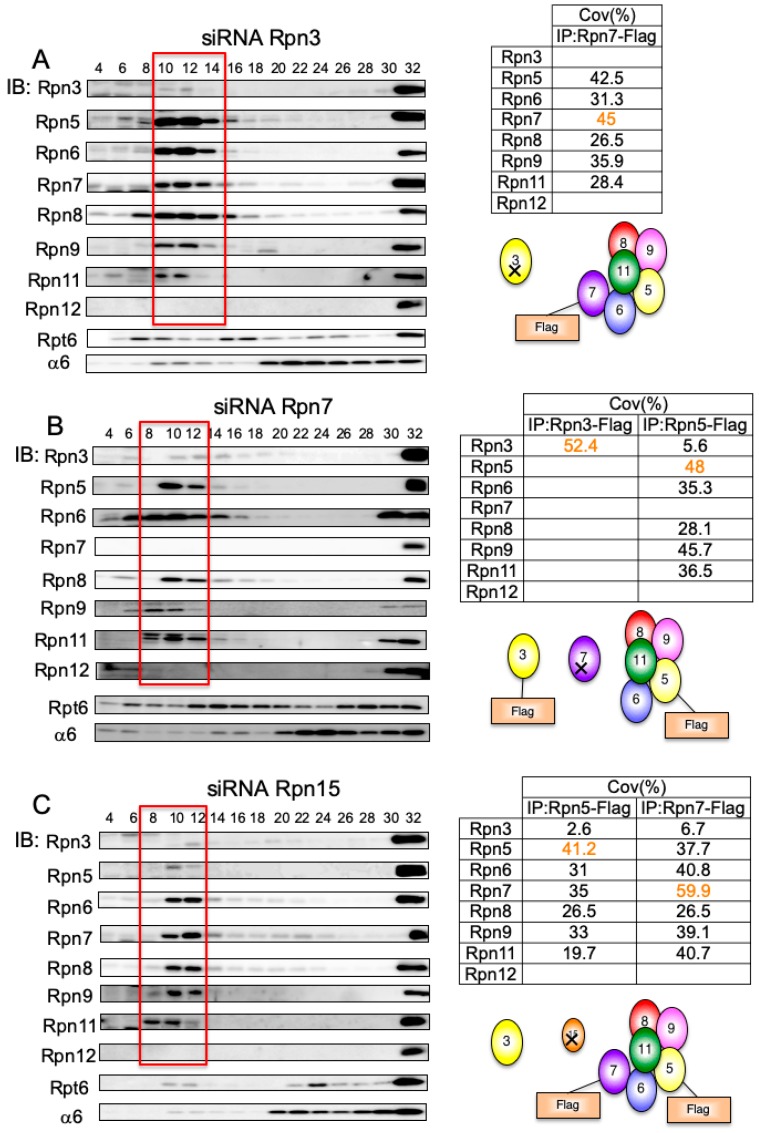
The Rpn7–Rpn6 interaction connects the Rpn3-7-15 module with the Rpn5-6-8-9-11 module. HEK293T cells stably expressing Rpn7-Flag treated with siRNA targeting Rpn3 (**A**), cells stably expressing Rpn3-Flag and Rpn5-Flag treated with siRNA targeting Rpn7 (**B**), and cells stably expressing Rpn5-Flag and Rpn7-Flag treated with siRNA targeting Rpn15 (**C**) were analyzed in the same way as described in Figure 2. The tables show the sequence coverage of the identified subunits.

**Figure 7 biomolecules-09-00213-f007:**
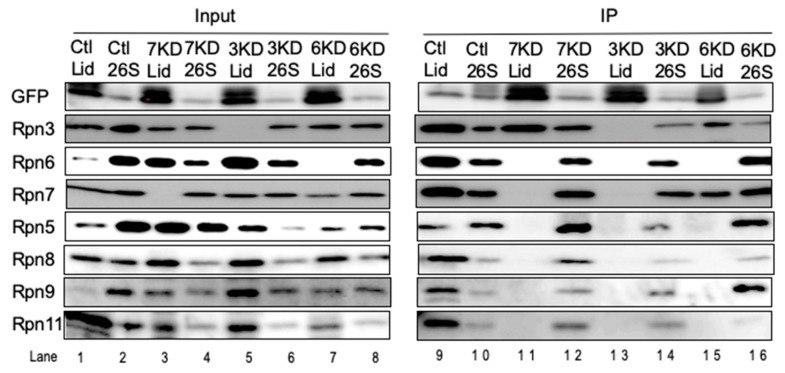
Rpn15 mediates the association of Rpn3 with Rpn7. siRNA targeting Rpn7, Rpn3, and Rpn6 was performed using HEK293T cells stably expressing Rpn15-GFP. Accumulated intermediates were collected and immunoprecipitated with anti-GFP antibody. When Rpn7 was knocked down, only Rpn3 was coimmunoprecipitated with Rpn15. None of the lid subunits was identified when Rpn3 was knocked down. Intermediates of Rpn3-7-15 were detected when Rpn6 was knocked down.

**Figure 8 biomolecules-09-00213-f008:**
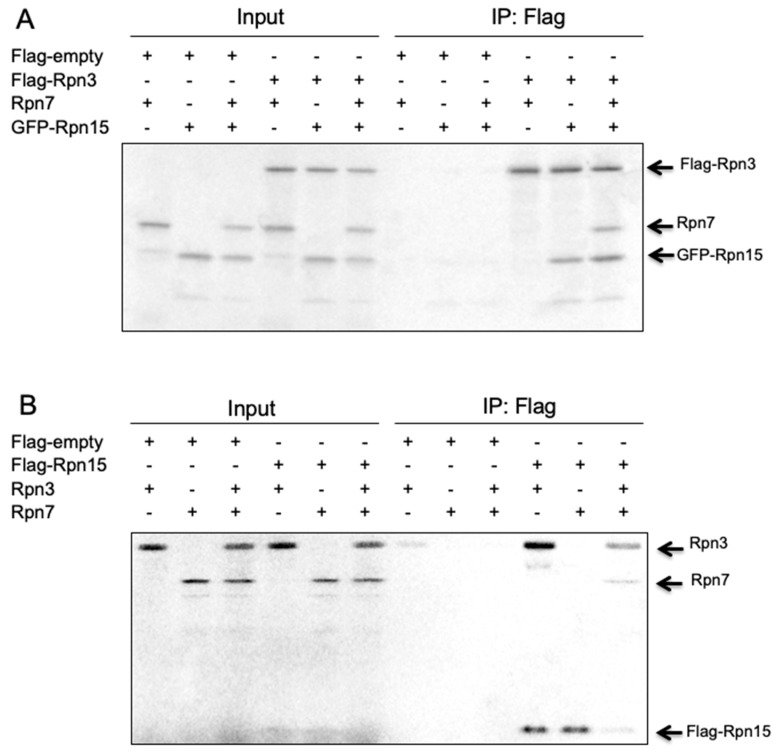
Rpn15 directly binds to Rpn3 and promotes Rpn3–Rpn7 association in vitro. Flag-Rpn3, Rpn7, and GFP-Rpn15 (**A**) or Flag-Rpn15, Rpn3, and Rpn7 (**B**) were cotranscribed/cotranslated and ^35^S-radiolabeled in reticulocyte lysates in various combinations as indicated. Anti-Flag agarose beads were added to the reaction mixture, and the immunoprecipitates were run on an SDS-PAGE gel. Coimmunoprecipitated proteins with Flag-Rpn3 (**A**) or Flag-Rpn15 (**B**) were visualized using autoradiography.

**Figure 9 biomolecules-09-00213-f009:**
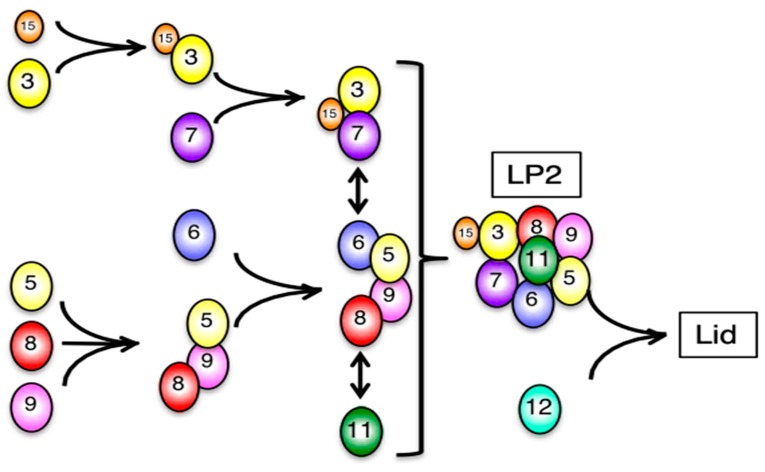
Assembly pathway of the mammalian proteasome lid subcomplex. The assembly of the lid subcomplex starts with the formation of two independent intermediates: Rpn3-15 and Rpn5-8-9. Then, Rpn7 and Rpn6 are incorporated into Rpn3-15 and Rpn5-8-9, forming Rpn3-7-15 and Rpn5-6-8-9, respectively. The Rpn5-6-8-9 complex is a prerequisite for incorporation of Rpn11. Rpn12 is the last subunit to be incorporated.

**Table 1 biomolecules-09-00213-t001:** RNAi sequences used in this study on the assembly pathway of the lid subcomplex.

Name	Sequence	Supplier
Human Rpn3	5′-UGUCCUGACAGCUUGAGUCAGAAGG-3′	Invitrogen
Human Rpn5	5′-UUCCAUAGUCCUCAACAAGUGUGGA-3′	Invitrogen
Human Rpn6	5′-UAGUAAGUUAUCAUACAACUUGGCC-3′	Invitrogen
Human Rpn7	5′-UAGUAAGGAGCCAUGUUGUUAUCGC-3′	Invitrogen
Human Rpn8	5′-UGUGGUACCAGCCAACUAUUCUUUC-3′	Invitrogen
Human Rpn9	5′-AAUUGUUUCCUUUGUAACCUGUAGG-3′	Invitrogen
Human Rpn11	5′-AUACCAACCAACAACCAUCUCCGGC-3′	Invitrogen
Human Rpn12	5′-UAUGUCACGGGCCAGAAUUAGCUGC-3′	Greiner bio-one
Human Rpn15	5′-UAACAGACCUAAGUCUACCGGCUGC-3′	Greiner bio-one

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
