# Peer review of "In-depth Analysis of the Lid Subunits Assembly Mechanism in Mammals"

_biomolecules, 2019, doi:10.3390/biom9060213_

Round 1

Reviewer 1 Report

The present work is a good dissection of proteasome lid assembly pathway in humans, which is similar to the pathway defined in yeast cells in previous works.

The research is well designed, performed and presented.

The experimental approach is based on human cell cultures transfected with appropriate vectors and a series of siRNAs in order to knock down the subunits of the lid. After cellular extracts were obtained, glycerol gradient centrifugations were carried out. Fractions obtained in the gradients were analyzed by western blot, and fractions containing the lid were further Flag-based inmunoprecipitated and analyzed by mass spectrometry. Using this approach the authors are able to recapitulate the pathway of lid assembly and to define the sub-complex hierarchy in the process of association.

minor points:

-Figure 6 , C. Why you don't detect Rpn5 in lid fractions (8-10-12)? 

-In the discussion, it  should be noted that the assembly process in humans is highly similar to the process defined by Tomko et al. Mol. Cell 2011, 44, 907‐917 (reference 24), corresponding to the distant eukaryote Saccharomyces cerevisiae. Moreover, the differences found between both species should be detailed.

Author Response

Reviewer 1

The present work is a good dissection of proteasome lid assembly pathway in humans, which is similar to the pathway defined in yeast cells in previous works.

The research is well designed, performed and presented.

The experimental approach is based on human cell cultures transfected with appropriate vectors and a series of siRNAs in order to knock down the subunits of the lid. After cellular extracts were obtained, glycerol gradient centrifugations were carried out. Fractions obtained in the gradients were analyzed by western blot, and fractions containing the lid were further Flagbased inmunoprecipitated and analyzed by mass spectrometry. Using this approach, the authors are able to recapitulate the pathway of lid assembly and to define the sub-complex hierarchy in the process of association.

Response: We thank the reviewer for favorable comments, and we are pleased to know that the reviewer positively evaluates our work.

minor points:

-Figure 6 , C. Why you don't detect Rpn5 in lid fractions (8-10-12)?

Response: Unfortunately, we have not yet known the exact reason. As described in the result, we speculate that it may be due to a potential protein modification (page 10, lines 249-251).

-In the discussion, it should be noted that the assembly process in humans is

highly similar to the process defined by Tomko et al. Mol. Cell 2011, 44, 907‐

917 (reference 24), corresponding to the distant eukaryote Saccharomyces

cerevisiae. Moreover, the differences found between both species should be

detailed.

Response: We appreciate this valuable indication. According to the reviewer comments, we modified the discussion regarding Rpn11 and Rpn15 (page 14, lines 333-335 and 341-350). 

Reviewer 2 Report

In the paper: In‐depth analysis of the lid subunits assembly mechanism in mammals contributed by: Minghui Bai, Xian Zhao, Kazutaka Sahara, Yuki Ohte, Yuko Hirano, Takeumi Kaneko, Hideki Yashiroda, and Shigeo Murata, the authors analyze in great detail the assembly of the subunits of the regulatory complex of the 26S proteasome. The applied methods, RNAi and mass spec, are state of the art, well described and appropriate to investigate the process of assembly. The results are presented in easy to understand figures, which can be understood without reading the whole text, and summarized in cartoons. The authors can present clear evidence the assembly starts with two sub complexes consisting either of Rpn 3 and 15 or Rpn 5,8 and 9. The first on integrates Rpn 7, the later one Rpn 6, the binding of Rpn11 joins the two sub complexes and the assembly is finished by the incorporation of Rpn12. The result and discussion sections are concise, but cover all results and stick to presented data and avoid lengthy may be discussions.

Author Response

Reviewer 2

In the paper: Indepth analysis of the lid subunits assembly mechanism in mammals contributed by:Minghui Bai, Xian Zhao, Kazutaka Sahara, Yuki Ohte, Yuko Hirano, Takeumi Kaneko, Hideki Yashiroda, and Shigeo Murata, the authors analyze in great detail the assembly of the subunits of the regulatory complex of the 26S proteasome. The applied methods, RNAi and mass spec, are state of the art, well described and appropriate to investigate the process of assembly. The results are presented in easy to understand figures, which can be understood without reading the whole text, and summarized in cartoons. The authors can present clear evidence the assembly starts with two sub complexes consisting either of Rpn 3 and 15 or Rpn 5,8 and 9. The first on integrates Rpn 7, the later one Rpn 6, the binding of Rpn11 joins the two sub complexes and the assembly is finished by the incorporation of Rpn12. The result and discussion sections are concise, but cover all results and stick to presented data and avoid lengthy may be discussions.

Response: We are glad to know that the reviewer understands our work well and evaluates it highly. 

Reviewer 3 Report

This contribution seeks to identify long lived proteasome lid subunit assembly intermediates using pulldowns of proteasomal subunits from human cells analyzed by blot and mass spectrometry. We concur with the authors that question is of high biological importance. While we do not recommend acceptance at this time, we could recommend acceptance of the publication following revisions to clarify the following points:

·         It is our opinion that the presented evidence for tight complexes is overstated in light of the results. This section would more accurately state that subunit abundances are correlated with regards to siRNA knockdown, and that a potential interpretation of this fact is the existence of the proposed complexes.

·         The authors should determine the stoichiometry of the subunits identified for each of the proposed complexes.

·         When possible, authors should obtain purified components and at least attempt assembly reactions of these intermediates to determine formation kinetics as well as potential sufficient conditions for formation.  If any part of purification is challenging, complementation experiments in cell lysate or IVT should be done at least.  The current work is alarmingly devoid of any evidence that those subcomplexes are indeed intermediates other than degradation fragments or assembly dead ends.

·         The authors should obtain coordinates from recent cryo-EM reconstructions of 26S proteasome complexes and prepare a figure showing positions of the lid subunits in the full complex and clearly indicate in this figure relationships between the assembled 26S proteasome and intermediates proposed from this study. It is our opinion that a terse statement that the two are consistent is insufficient given the ready accessibility of these coordinates for comparison.

·         The authors should clearly state whether, in mass spectrometry analysis, other proteins were coidentified with proposed complexes aside from proteasomal subunits.

·         The authors should include in their analysis whether they checked for/identified post translational modifications in peptides identified from proteasomal subunits and assigned to complexes. The coverage presented is rarely complete.

·         The authors state that Rpn15 produces suboptimal peptides and is difficult to blot, and generate a GFP-Rpn15 construct. It is not clear how the authors verified that appending GFP to Rpn15 does not alter its function, and more importantly, its role in the lid assembly pathway.

·         The authors should more clearly state why peptides from Rpn15 are suboptimal for mass spectrometry, and whether there are significant differences from other subunits.

Minor point:

The author said “..location of Rpn15 is not assigned in the cryoelectron microscopy structure…”. This is not accurate. Sem1 has been identified and traced, less a small segment, in most recent cryo-EM structures of human 26S proteasome.

Author Response

Reviewer 3

This contribution seeks to identify long lived proteasome lid subunit assembly intermediates using pulldowns of proteasomal subunits from human cells analyzed by blot and mass spectrometry. We concur with the authors that question is of high biological importance. While we do not recommend acceptance at this time, we could recommend acceptance of the publication following revisions to clarify the following points:

Response: We appreciate the critical comments by the reviewer 3. These comments give us an opportunity to reconsider our results and improve our manuscript. 

· It is our opinion that the presented evidence for tight complexes is overstated in light of the results. This section would more accurately state that subunit abundances are correlated with regards to siRNA knockdown, and that a potential interpretation of this fact is the existence of the proposed complexes.

Response: We appreciate the thoughtful comment. According to this comment, we modified the text in the introduction (page 2, lines 66-69). 

· The authors should determine the stoichiometry of the subunits identified

for each of the proposed complexes.

Response: We appreciate this comment. The experiment suggested by the reviewer would provide us detailed information on each of the complexes purified in this study. However, we have already had a reliable lid assembly model from yeast studies, and therefore we can make a mammalian model based on our results, whether they included complexes with abnormal stoichiometries or not. Therefore, we don’t think it is indispensable to determine the stoichiometry of the subunits in each intermediate. However, we admit that we cannot exclude the possibility that the intermediates we examined are non-physiological intermediates with abnormal subunit stoichiometries and dead-end products. We discussed this point in the discussion section (page 14, lines 325-330). We would appreciate it if the reviewer could reserve this topic. 

·When possible, authors should obtain purified components and at least attempt assembly reactions of these intermediates to determine formation kinetics as well as potential sufficient conditions for formation. If any part of purification is challenging, complementation experiments in cell lysate or IVT should be done at least. The current work is alarmingly devoid of any evidence that those subcomplexes are indeed intermediates other than degradation fragments or assembly dead ends.

Response: We admit that the experiments suggested by the reviewer would be of help to confirm our lid assembly model. However, we can propose our model without assuming degradation fragments and assembly dead-end products regarding the intermediates we examined. We would greatly appreciate it if the reviewer could exempt in vitro experiments in this manuscript. 

·The authors should obtain coordinates from recent cryo-EM reconstructions of 26S proteasome complexes and prepare a figure showing positions of the lid subunits in the full complex and clearly indicate in this figure relationships between the assembled 26S proteasome and intermediates proposed from this study. It is our opinion that a terse statement that the two are consistent is insufficient given the ready accessibility of these coordinates for comparison.

Response: We did not intend to use the word “consistent”, taking into consideration the coordinates of the 26S proteasome and the intermediates. Accordingly, we deleted the corresponding sentence in the discussion section (page 14, line 340). 

· The authors should clearly state whether, in mass spectrometry analysis, other proteins were coidentified with proposed complexes aside from proteasomal subunits.

Response: We have detected proteins other than lid subunits in the intermediate complexes, but we have not yet discriminated whether these proteins are merely contaminated sediments or have some biological importance for the lid assembly. Accordingly, we withdraw our claim that no chaperone seems to be required for the lid assembly in the abstract (page 1, line 22) and the conclusions (page 15, lines 535). We also added and modified sentences regarding this point in the discussion (page 14, lines 363-368). 

· The authors should include in their analysis whether they checked for/identified post translational modifications in peptides identified from proteasomal subunits and assigned to complexes. The coverage presented is rarely complete.

Response: We have not performed mass spectrometry analyses to specifically identify post-translational modifications. The coverage of peptides depends on binding and elution efficiency of the column we used for liquid chromatography, the nature of each peptide and its compatibility with ionizing method (in our case, MALDI), and the sensitivity of the instrument. Therefore, it is usual that the coverage is below 100% even when we analyze purified proteins in a sufficient amount, at least in our experimental condition.

· The authors state that Rpn15 produces suboptimal peptides and is difficult to blot, and generate a GFP-Rpn15 construct. It is not clear how the authors verified that appending GFP to Rpn15 does not alter its function, and more importantly, its role in the lid assembly pathway.

Response: We admit that we have not yet proved that appending GFP to Rpn15 does not alter its function. The previous study (Tomko, R.J. and Hochstrasser, M., Mol. Cell., 53, 433-443, 2014) showed that Sem1 sustains its native ability to bind to both Rpn3 and Rpn7 even when it was N- and C-terminally tagged with FLAG and the ZZ-6His domain. Since structures and functions of the proteasome are closely similar between mammals and yeast, we believe that GFP-Rpn15 does not alter its function.

· The authors should more clearly state why peptides from Rpn15 are suboptimal for mass spectrometry, and whether there are significant differences from other subunits.

Response: Unfortunately, we have not yet known the exact reason. We examined the content of lysine and arginine in Rpn15, which are critical residues for trypsin recognition. However, there is no significant difference between Rpn15 and the other subunits. Probably, it is due to binding and elution efficiency of the column we used for liquid chromatography, the nature of peptides and its compatibility with ionizing method.

Minor point:

The author said “..location of Rpn15 is not assigned in the cryoelectron microscopy structure…”. This is not accurate. Sem1 has been identified and traced, less a small segment, in most recent cryo-EM structures of human 26S proteasome.

Response: We sincerely apologize for not referring to the advance on Sem1 published recently. Thanks to this valuable comment, we reinterpreted our results regarding Rpn15 and modified the text in the result (page 13, line 301) and discussion (page 14, lines 341-350) and figure 9.

Round 2

Reviewer 3 Report

I appreciate the revision by the author. Most of my questions have been addressed. I can now recommend it for publication.